# The Experience of Prisoners with Serious Mental Disorders Participating in a Dog-Assisted Therapy Program: A Qualitative Study

**DOI:** 10.3390/ani15030379

**Published:** 2025-01-28

**Authors:** Nuria Máximo-Bocanegra, Domingo Palacios-Ceña, Javier Güeita-Rodríguez, Sergio Serrada-Tejeda, Ana García-Medranda, Alba Pacheco-Guijarro, Carlos Pacheca-Flores, Jorge Pérez-Corrales

**Affiliations:** 1Animals and Society Chair Program, Universidad Rey Juan Carlos, 28922 Alcorcón, Spain; nuria.maximo@urjc.es; 2Research Group in Evaluation and Assessment of Ability, Functionality and Disability of Universidad Rey Juan Carlos (TO+IDI), Department of Physical Therapy, Occupational Therapy, Rehabilitation and Physical Medicine, Universidad Rey Juan Carlos, 28922 Alcorcón, Spain; 3Research Group of Humanities and Qualitative Research in Health Science of Universidad Rey Juan Carlos (Hum&QRinHS), Department of Physical Therapy, Occupational Therapy, Rehabilitation and Physical Medicine, Universidad Rey Juan Carlos, 28922 Alcorcón, Spain; domingo.palacios@urjc.es (D.P.-C.); javier.gueita@urjc.es (J.G.-R.); jorge.perez@urjc.es (J.P.-C.); 4El Dueso Penitentiary Center, 39749 Santoña, Spain; 5Fundación Madre, 19003 Guadalajara, Spain; 6Community Social Support Team, 15940 Barbanza, Spain

**Keywords:** mental disorders, dog-assisted therapy, prison, qualitative research

## Abstract

Prisoners with serious mental health conditions often face significant challenges in their rehabilitation and emotional well-being. This study explored the effects of a dog-assisted therapy program in a Spanish prison, focusing on its impact on emotional health and social connections. Over two months, sixteen male prisoners participated in weekly sessions with a trained therapy dog, guided by professional therapists. The program aimed to reduce isolation, improve mood, and foster trust. Participants reported feeling happier and less lonely, often describing the therapy sessions as the highlight of their week. They emphasized the unconditional affection they received from the dog, which provided emotional comfort, and a sense of connection that is rarely found in prison. The bond with the dog also helped them reflect on their relationships with others and themselves, encouraging trust and self-awareness. Participants valued the supportive role of the therapists, who provided guidance and emotional support during the sessions. Overall, the program was described as a positive, transformative experience, promoting personal growth and emotional resilience. This study highlights the potential for therapy programs involving animals to improve mental health and enhance rehabilitation in correctional settings, suggesting these programs could encourage wider efforts to support recovery and reintegration into society.

## 1. Introduction

Effective development of policies, programs, and services that improve health outcomes for inmates with a severe mental disorder (SMD) requires using “their lived experiences as a form of consciousness” [1]. This approach is consistent with the principles of the Recovery Model, which emphasizes the importance of connection, hope, identity, meaning, and strength in the recovery process for people with mental health conditions [2,3]. Although the recovery model is widely used in community mental health settings, its application in correctional settings remains limited [4]. Nevertheless, inmates with mental health conditions acknowledge these principles as essential elements for their own recovery [5]. This context underscores the need for interventions that incorporate these elements into prison-based mental health care.

People with mental health conditions and substance use disorders are highly represented in the prison context, a trend observed worldwide. In this regard, different research has identified that the most frequent rate is associated with major depression, psychosis [6,7], and substance use disorders [8,9]. In this regard, the prevalence of SMD among prisoners is of particular concern. Researchers Fasel and Seewald [7] indicated that 3.6% of male and 3.9% of female prisoners had a mental health condition, rising to 6.2% in low- and middle-income countries, especially in regions such as Africa, South America, and Southeast Asia [10]. Mental health conditions associated with SMD (or Severe Mental Illness) include people with non-organic psychotic disorders such as schizophrenia and related psychotic disorders, personality disorders with psychotic features, of long duration (≥2 years), who present with a variable degree of disability and social, occupational and/or work dysfunction, and who must be cared for in different devices of the psychiatric and social health care services [6,11]. In Spain, the prevalence of SMD among prisoners is 4.3%, more than double that observed in the general population [12]. These disparities underline the urgent need for targeted and effective mental health interventions in prisons.

In this sense, criminal justice policies seek to address the mental health needs of prisoners, especially in low- and middle-income countries through cost-effective and scalable interventions that support the achievement of these goals [10]. In Spain, Peiró [13] highlights the importance of improving mental health care in prisons and advocates for prioritizing cost-effective interventions within criminal justice policies. International perspectives further underscore the potential benefits of prison-based interventions. For example, studies conducted in the United States [14], Canada [15], and Italy [16] demonstrate that prison programs focused on developing empathy, respect, and trust have positive effects on mental health and rehabilitation outcomes for inmates. In this regard, the quality of care for prisoners with mental health conditions is a priority [17], and it has been identified as an urgent need to incorporate psychosocial interventions in the prison setting [18].

One such intervention is animal-assisted intervention (AAI), conceptualized by the International Association of Human–Animal Interaction Organizations (IAHAIO) [19] as an approach that uses animals to achieve therapeutic, educational, or social benefits in the population. In this context, animal-assisted therapy (AAT), considered a type of AAI, is a therapeutic intervention led by a health, educational, or social professional with the aim of achieving physical, cognitive, emotional, or social improvements. In relation to this, current evidence suggests that dog-assisted therapy (DAT), a subtype of AAT, can provide significant benefits in populations with mental health conditions, such as a reduction in anxious and depressive symptomatology or improvement in self-esteem and quality of life [20,21]. These benefits are particularly promising in the prison context, as AAI programs try to harness the human–animal bond to improve the well-being of inmates. In this sense, early experiences with AAI in U.S. correctional facilities have shown reductions in violence, suicide attempts, and medication dependency among inmates who participated in programs involving dogs, horses, and farm animals [22].

However, most existing studies have methodological limitations, and the outcomes are highly variable depending on the type of intervention, the characteristics of the participants, and the program’s design [23]. Despite this, AAI programs are now implemented in correctional systems worldwide, mainly designed to improve the rehabilitation of inmates, encourage responsibility, empathy, and social skills as well as show improvements in inmates’ emotional well-being and a reduction in aggression [24]. In Spain, Villafaina-Domínguez et al. (2020) [23] conducted a systematic review concluding that this type of program can improve several variables, such as mental health, emotional control, empathy, and academic skills in both male and female prisoners. However, the included studies were of suboptimal methodological quality, and the participants and results were highly heterogeneous, including programs which have typically involved either visits from therapy dogs or direct interaction with rescued dogs through care and training activities [25,26,27].

On this basis, several studies have highlighted the therapeutic potential of animal-assisted interventions (AAIs) in prison settings, particularly for prisoners with psychiatric disorders. In this sense, Mercer et al. (2015) [28] explored the experiences of psychiatric prisoners participating in a DAT, finding significant emotional benefits such as reduced anxiety, improved self-esteem, and a stronger sense of emotional connection. Additionally, inmates gained a sense of responsibility and accomplishment by caring for the dogs, which contributed to their rehabilitation and social skills development. Similarly, Mulcahy and Mclaughlin (2013) [29] have examined the effectiveness of AAI programs in prisons, focusing on the psychological benefits for prisoners and highlighting the positive emotional, psychological, and social effects for incarcerated individuals. These benefits include stress reduction, reduced levels of anxiety and depression, increased empathy, and improved emotional regulation. In addition, they allow for the development of social skills and promote the potential for rehabilitation through increased responsibility, care, and bonding with animals. Similarly, evidence suggest that participating in a DAT showed that the therapeutic relationship with the animal fostered empathy, self-regulation, and a sense of responsibility, contributing to the rehabilitation process of the inmates [30]. These characteristics support AAI being considered as a component of a comprehensive mental health and rehabilitation framework for prisoners.

However, and although research on AAI for general prison populations has grown, there is a notable gap in knowledge regarding the specific experiences and outcomes for prisoners with SMD. Therefore, this study seeks to address this gap by exploring the experiences of Spanish prisoners with SMD who participate in a DAT program. By focusing on this specific population, this study aims to provide new insights into the potential benefits of DAT for incarcerated individuals with mental health conditions. This knowledge could inform the development of more effective, evidence-based mental health interventions within the Spanish prison system and beyond.

## 2. Materials and Methods

### 2.1. Study Design

A descriptive qualitative study was conducted based on an interpretive framework. This paradigm describes how people construct their own social reality and knowledge through reconstructions of their particular human experiences and their contexts, as recognized from different subject positions [31]. The Standards for Reporting Qualitative Research (SRQR) [32] and the Consolidated Criteria for Reporting Qualitative Research (COREQ) [33] were followed. Qualitative studies deepen the understanding of the perceptions, emotions, beliefs, and motivations of people who experience complex phenomena from their own perspective [34,35]. Qualitative research has been used in the study of the experiences of people with SMD, proving useful for understanding the meaning that people attribute to their own experiences of recovery [3]. Jalongo [36] emphasizes the importance of personal narratives when studying DAT programs in prisons, highlighting how these narratives enable an in-depth exploration of the lived experiences of inmates. Jalongo [36] argues that, supported by recent findings in neuroscience, the use of a narrative can have a significant impact on rehabilitation and the emotional well-being of participants. Additionally, Jalongo [36] maintains that personal stories provide valuable insights that quantitative methods alone might not capture, thus justifying the use of qualitative methodologies to capture the complexity and human impact of these interventions. This approach underscores the legitimacy and value of subjective experiences in psychosocial and correctional research [36].

This study was approved by the Research Ethics Committee of the Rey Juan Carlos University (Code: 2104202112221) and by the General Directorate of Penitentiary Institutions of the Spanish Ministry of the Interior. The DAT program in this study adhered to the animal welfare standards set by the Institutional Chair of Animal and Society Research at Rey Juan Carlos University. This organization ensures the ethical selection of human–animal teams and the commitment to ethical, rigorous, and empathetic work by all parties involved.

The study obtained special permission from the General Secretariat of Penitentiary Institutions to involve incarcerated individuals, emphasizing that participation had to be both voluntary and anonymous. Participants received detailed information about the study, which outlined its voluntary nature and the importance of informed consent. After being fully briefed, they signed the consent forms. To ensure confidentiality, the researcher conducting the interviews did not have access to the participants’ personal data or the reasons for their incarceration, and the team responsible for the DAT sessions was not present during the interviews. Conducting research in a prison context, and with individuals with SMD, poses significant ethical challenges, including addressing participants’ vulnerability and ensuring informed consent is adequately obtained. The research team managed these challenges by implementing strategies such as strict confidentiality measures, using clear and understandable language, and providing ongoing supervision and support throughout the study.

The URJC Ethics Committee, that reviewed the project, has requirements for studies involving animal-assisted interventions. These include a review of the health status of the animals and their physical and emotional well-being, as well as liability insurance. All these processes were completed to enable the implementation of the study. In terms of dog welfare, the guidelines of the IAHAIO [19], as well as those of Johnson et al. (2019) [37], were considered. These guidelines refer to compassionate and respectful training methods for dogs without any type of exploitation. Dog handlers were also required to be trained in canine communication, including animal body language and stress signals, as well as basic dog behavior and non-force training techniques.

### 2.2. Context

This study was conducted at the Navalcarnero–Madrid IV Prison (Madrid, Spain) among participants with SMD who participated in the DAT program. The participants in the program were people deprived of their liberty who participated in the Psychosocial Care and Community Intermediation Service (SAPIC).

A more detailed description of the context can be found in the Appendix A [19,36], as well as the description of the activities carried out in the Appendix A.

### 2.3. Participants and Sampling Strategies

Purposive sampling was used [35,38] to include participants with relevant information (prisoners who participated in the DAT program). All prisoners with SMD who participated in the DAT program were included in this study.

The inclusion criteria were (a) inmates of the Navalcarnero–Madrid IV Prison; (b) aged ≥18 years; (c) diagnosed with SMD; (d) participating in the DAT program; and (e) accepting and signing the informed consent form. The exclusion criteria were (a) acute psychopathological instability; (b) difficulties affecting understanding or communication; and (c) refusal to sign the informed consent.

### 2.4. Procedure

Data collection took place over the months of June and July 2021. In-depth interviews were conducted, and field notes were taken [35,38]. A semi-structured question guide was used (Table 1). The questions were open-ended to allow the participants to unrestrictedly narrate their experience and to obtain an in-depth description from their perspective [35]. A total of 736 min of interviews were recorded. The average duration of the interviews was 46 min, and they were conducted face-to-face in a room provided by the prison. Only the researcher (JP-C) and the participant were present during the interviews. All interviews were audio-recorded, with the consent of the participants and the prison.

### 2.5. Analysis

An inductive thematic analysis was applied [30]. Each interview was transcribed (NM-B, AG-M, AP-G, CP-F). Subsequently, the text fragments that provided relevant information for the study were identified. To conceptualize “relevance” in the thematic analysis, the researchers responsible for analyzing the results (JP-C and JG-R) identified text fragments that directly aligned with the study’s research objectives, which were defined in relation to the chosen theoretical framework. Specifically, we considered relevant any data illustrating participants’ experiences, perceptions, or interpretations of participating in the dog-assisted intervention program. The active selection of relevant data was guided by our interpretative lens and expertise in the process of recovery for individuals with SMD [3], ensuring that the coding process remained theoretically grounded and contextually appropriate. From these fragments, narratives were obtained and coded (identification of key contents). The codes were then grouped into clusters (categories) with common contents. Finaly, themes were obtained from the grouping of the categories. The data analysis process has been explained in greater detail in Appendix A. The results are presented with examples of participants’ narratives to ensure the traceability of the results [35,39], both in Section 3 and in Appendix A. Double independent coding was applied, performed by JG-R and JP-C. Subsequent meetings were necessary to combine the results of the prisoners’ experience, reaching a consensus on the establishment of the themes. No qualitative software was applied for the data analysis.

For the trustworthiness of the data, the Lincoln and Guba criteria were followed for credibility, transferability, dependability, and confirmability [40]. Triangulation of researchers and data collection instruments was performed during the analysis, together with member checking, complete description of the study, a reflexivity process, and description of tree coding from the narratives to the final themes.

Reflexivity was established through preliminary meetings among the researchers, based on how the DAT program sessions had been conducted. The initial researcher positioning assumed that this type of intervention could be beneficial for the participants, which also highlighted the need to ask participants about potential negative aspects of their participation. Member checking was conducted after the collection of primary data, with the researcher summarizing their interpretation of the data gathered at the end of each interview with the participants [41]. This approach aimed to contrast the researcher’s impressions with those of the participants. For example, when presenting the data related to the emotional impact of participating in the DAT program—such as feelings of joy and happiness or the importance of the bond with the dog—participants were able to confirm whether their experiences were accurately represented or provide additional nuances. This process ensured that the selected quotes and interpretations faithfully reflected their perceptions, avoiding misunderstandings or distortions in the results. Furthermore, participants’ feedback during this process enhanced the credibility of the data by incorporating their own words and perspectives into the final analysis.

## 3. Results

Of the 17 prisoners who participated in the DAT program, 16 were recruited for the present study. One participant initially agreed to participate but later withdrew due to a lack of motivation after several unsuccessful attempts to conduct an in-depth interview. All participants had a diagnosis of SMD, with a mean age of 43.4 years (SD ± 11.4) and a sentence time of 5.9 years (SD ± 5.6). This study was conducted in a male prison, therefore no women participated. Sociodemographic and clinical data are shown in Table 2.

Two themes emerged: (a) the emotional impact of participating in the DAT program, with three categories: experiencing positive feelings, bonding with the dog, and feeling helped by the therapists guiding the sessions; and (b) the process of participating in the DAT program, with three categories: the decision to participate in the program, initial disbelief and surprise, and progressive changes in their experience.

### 3.1. Theme 1: The Emotional Impact of Participating in the DAT Program

#### 3.1.1. Experiencing Positive Feelings

All participants related that participating in the program was very positive, which generated feelings of joy and happiness from being in contact with the animals. The participants described how it helped them to improve their state of mind inside the prison and thanked the prison for giving them the opportunity to participate in the program. *“Before you even walk through the door, the animal comes out to see you, recognizes you and all of this is gratifying. It brings you joy, and you see how the animal rejoices too, as soon as she sees you. Anything here is amplified, it’s something special.”* (Participant 3); *“The mood, maybe I was feeling cross, but I got over it, I was feeling bad for whatever reason, and then I came out better. Very gratifying”.* (Participant 6).

The participants described that the feelings generated by their participation in the program are passed on or “spread” by the dog itself, leading them to experience the feelings of joy and happiness shown by the dog. In addition, the feelings experienced during the DAT sessions were transferred to their day-to-day life in prison and were maintained over time. *“The dog is happy and content too, and this is contagious. The beauty of the animal, how sweet she is, the fur, how well cared for the animal is... all of that gives you happiness.”* (Participant 16); *“You spend the week better, you’re happier in the module and you know that next week she comes again [the dog] (…) when I’m going to go to our session, I say to myself <<I’m going to see the animals, what fun!>>”* (Participant 11).

#### 3.1.2. Bonding with the Dog

Participants described that what they valued most was the bond they established with the dog. This bonding was possible because of the unconditional love and affection they felt from the dog. Some participants highlighted that the dog did not judge them, feeling more comfortable with the animals than with many people. In addition, participants recounted how the affection was reciprocal, they also felt great love and affection towards the dog. The interviews reflected that what they valued most was the bond they established with the dog. This bonding was possible because of the unconditional love and affection they felt from the dog. Some participants highlighted how the dog does not judge them, feeling more comfortable with the animals than with many people. In addition, participants recounted how the affection was reciprocal, they also felt great love and affection towards the dog. A striking finding was that participants reported that being able to pet and give affection to dog is a way of giving and feeling affection for themselves. *“It’s a bond of unconditional, selfless love, and that’s not easy to find in normal life. It makes me feel affection for the animal and that affection makes me feel good towards myself (…) the dog does not judge you, and seeing how the animal behaves with you, it gives you an understanding of how a person should behave with another person, it is much more pure and healthy.”* (Participant 2); *“You have become an unwanted person, so a token of affection, no matter if it is from an animal, is priceless and has a psychological and emotional impact”.* (Participant 3).

Participants recounted how these displays of affection in prison are important to them, because in prison they are away from their loved ones and feel lonely and lacking affection. Some described how they even isolate themselves from other prisoners because of the harmful relationships that exist within the prison. Thus, they placed great value on the bond and affection that the dog showed towards them, which had a positive impact on their daily life in prison. *“There are many people here who only think about how to get money out of you and that makes you isolate yourself; however, you counteract that negativity with the presence of an animal that is faithful, that gives you affection.”* (Participant 9); *“In prison we have shortcomings, and there is so much love, harmony, and respect that we also internalize it.”* (Participant 14); *“Here we are lacking in affection. Affection here is difficult, who gives us affection? Affection is given to us by our family, our loved ones, our animals. The little dog has given me much affection.”* (Participant 16).

#### 3.1.3. Feeling Helped by the Therapists Guiding the Sessions

In addition to the bond with the dog, the relationship they have established with the therapists in charge of the therapy has been of great importance to their process. The participants described the therapists as professionals who showed concern towards them, feeling listened to, helped, and accompanied. Participants placed great value on the relationship they built, based on closeness and trust, feeling supported when they needed it. *“Both regarding the animal and the therapists who provide the therapy, because it strengthens that bond. The relationship has always been very good, close, and pure trust.”* (Participant 1); *“With them I have always had a good feeling, empathy, and reciprocity, we have understood each other very well. I have felt listened to, helped, and it’s one more thing to be grateful for.”* (Participant 3).

### 3.2. Theme 2: The Process of Participating in the DAT Program

#### 3.2.1. The Decision to Participate in the Program

The participants described their previous affinity for animals, specifically dogs, as the main reason for participating in the program. They experienced it as an opportunity given to them from prison. Other reasons participants gave were to keep busy in prison or because they felt involved with the service for people with SMD. *“Because I have always liked dogs. I haven’t petted a puppy for eleven years, and the truth is, I’ve really enjoyed it.”* (Participant 9); *“Just to avoid being here... sitting around, you have to go to school and go to programs and do things, so you aren’t thinking about anything else because if not, being in prison is more difficult, it’s better to be occupied.”* (Participant 8).

Lastly, there were also participants who chose to join the program for the opportunity to experience new things and learn: *“It has opened doors to things I don’t know, to therapies I’m not familiar with yet, and that I’m willing to try to see how it goes, what the experience is like... driven by the desire to learn new things.”* (Participant 1).

#### 3.2.2. From Initial Surprise and Disbelief to the Need for Incorporation as Therapy

The participants described how they never believed the animal program would take place, due to the security conditions in the prison. This feeling of surprise was widespread. Before the program, there were participants who identified the entry of dogs in prison with the objective of finding prohibited substances in the modules, not perceiving the animal as a therapeutic tool. *“The first session I even got a little excited, I couldn’t believe it, I said <<wow, how amazing!>>, in the end, they let them bring in the dog.”* (Participant 15); *“I thought it was a joke that you were going to bring the dog; I didn’t believe it. The dogs that come here are to see if people are bringing drugs. When I saw the dog, I said <<oh my God, so it’s real!>> My hair stood on end.”* (Participant 16).

However, the participants agreed on the need to incorporate this type of program into the therapeutic activities of prisons, allowing participation not only for inmates at their own facility but also for those in other penitentiaries: *“They should seriously consider implementing it and making it available in all facilities. They should introduce it because, as I said, it’s a release valve, especially for the participants.”* (Participant 1); *“I think it’s very positive, and I believe many people [inmates from other prisons] should experience what my companions and I have felt in these sessions, because it broadens one’s perspectives on life and everything.”* (Participant 2).

#### 3.2.3. Progressive Changes in Their Experience

The participants reported changes since they started participating in the program, from the first to the last session. Some participants described how, at the beginning, they felt more insecure with the animal because they did not know how to treat it. Others described how their motivation increased as they participated in the sessions, feeling better and more active. There were participants who perceived a great change, since at first, they were reluctant to bond with the animals because they did not want to feel affection or positive emotions inside the prison.

There were also participants who had owned dogs or other animals prior to their entry in prison, and others who intend to do so when they are released from prison. Among these participants, some described the first session as very emotional for this reason, and others thought they might feel bad about having memories from outside prison.

Finally, most of the participants recounted that, as the program progressed, they felt more confident with the dog, with the therapists, with their peers, and with themselves. *“At the beginning I was not very eager, very willing, but as the sessions went by, I was looking forward to the day, because the truth is that it gives you an extra something.”* (Participant 3); *“At the beginning I was more self-conscious and then I became more confident, both with the animal and with myself, my colleagues and the group.”* (Participant 6).

## 4. Discussion

Most of the programs developed in prisons involving dogs aim to rehabilitate inmates both behaviorally and emotionally. They are based on the training of dogs for adoption or service, with the aim of reducing recidivism. In Spanish prisons, there is a high prevalence of severe mental disorders and insufficient community resources are available. Despite the existence of specialized programs and mental health units within this context, different circumstances, such as overcrowding, or the lack of specialized training among prison staff, often compromise the effectiveness of the intervention. Prisoners usually receive psychological treatment and occupational therapy, which significantly influences their participation in therapeutic programs. In the present study, a specific program was designed for prisoners with SMD who are serving a sentence in an ordinary prison. In this sense, the proposed study aims to find out the impact and the process of participation in a specially designed program from a therapeutic point of view (DAT), which included the pairing of dogs and their guides specially prepared for this type of profile (male prison population with mental illness), accompanied by an expert in animal-assisted intervention with a professional occupational therapy profile. This program integrates activities with a specially trained dog into the occupational therapy (OT) process, with the aim of evaluating the inmates’ lived experience. This aspect is crucial to understand their perception of the illness and the treatment they receive in the prison context, improve clinical outcomes by encouraging adherence to OT sessions, strengthen the relationship of trust between the person and the health professional, help identify non-obvious needs, and humanize care while respecting the patient’s dignity. However, although efforts have been made to improve care and treatment, addressing these issues effectively requires continuous attention and adequate resources.

Following this approach, Allisson and Ramaswamy [14] presented a proposal to adapt animal-assisted therapy programs in prisons, recognizing that these programs could help inmates without diagnosed pathology to improve their physical and mental health (empathy, self-awareness, self-esteem) and reduce isolation, aiming to rehabilitate inmates behaviorally and emotionally. Duindam et al. [42] have shown that intervention programs, including prison dog training programs, not only reduce recidivism, but also significantly improve the emotional and social functioning of inmates. In addition, they provide mutual benefits, improving the well-being of both inmates and dogs, and act as catalysts for a more humane and constructive prison environment. However, although there are a growing number of studies on dog-assisted interventions in the prison population, the variability in methodologies, target populations and objectives make it difficult to establish clear benefits [23].

In 2019, similarly to our study, Dell et al. [15] evaluated the effect of dog-assisted therapy in a Canadian psychiatric correctional facility, finding similar results to those presented in this paper. The inmates’ connection with the dogs, through the perception of love and support from the animals, helped with the recognition of their personal feelings and emotions, positively improving their behavior. The results of our study provide insight into the impact and process of participating in this type of therapy. Participants’ experiences of their participation were positive, especially in terms of their feelings and ability to bond with the animal, the support received from the therapists, and the perceived change in their emotions and daily life within the prison during the program. Cooke and Farrington [25] and Minton et al. [43], in their work on the use of a DAT for women in prison, showed positive effects in the emotional domain. Arkow [44] described how animals can trigger happy memories, improve mood, and provoke a sense of happiness, joy, and general well-being.

Bonding is a key element in animal-assisted intervention programs [30], based on attachment theory [45]. Therefore, it was necessary to investigate this aspect in the prison context. Participants in our study highlighted the positive impact of the attachment and unconditional love received by the dog, Ara, on their mood. Smith et al. [30] specifically studied the bond between psychiatric prisoners and therapy dogs, finding that security, physical contact, reciprocity, and acceptance were relevant factors for the participants. The findings of this work suggest that therapy dogs may act as surrogate attachment figures for prisoners, mitigating their experiences of disconnection and fostering the development of interpersonal connections. Aligned with these findings are those of Leonardi et al. [46] who showed how prisoners who participated in these programs perceived this unconditional love and affection, which improved their mood, well-being, motivation, self-efficacy, impulse control, and emotional management. Furthermore, the work of Dell et al. [15] reports that the unconditional love and support of dogs was also described by the participants in their study, inmates feel disconnected from love and affection while incarcerated, considering this bond with the animal to be valuable. This connection with the dog was already described by mentally ill prisoners in Jasperson’s study [26], where they expressed that, thanks to this bond, they felt more willing to participate in group conversations and activities. Furthermore, the female prisoners in Cooke and Farrington’s study [25] identified the DAT as a safe space where trust was fostered.

Regarding the participation in our study and its impact on the daily life of the prisoners, it was found that their perception in this respect was very positive and much better than they had originally thought. Not only regarding the dog, but also with regard to the view they had of their colleagues and professionals. A positive prison climate that cultivates trusting therapeutic relationships can significantly reduce dynamic risk factors among inmates, such as hostility, impulse control, and criminal attitudes, underlining the importance of relational factors in the treatment of inmates [47]. Furthermore, a meta-analysis by Yoon et al. [48] demonstrated the benefits of a positive relationship and therapeutic bonding between therapists and prisoners, markedly improving treatment outcomes for depression and anxiety. The improvement in the relationship between participants in these programs and the professionals guiding the DAT appears in the work of Minton et al. [43] and Minke [49], where inmates used the term ‘normalizing’ the prison through the DAT. Minke [49] showed how DAT improved social relationships between inmates and between the staff running the sessions and the inmates; however, the improved relationships did not carry over to the prison staff. Similarly, Minton et al. [43] described how inmates felt helped by the therapists leading the sessions. Smith and Smith [50] detail how participants identified with the emotions of the dogs, perceiving a symbiotic relationship that fostered the well-being of them both and the relationship with the professionals accompanying them in the process.

Finally, no studies have been found that analyze the changes experienced in the experience of progressive participation in this type of program. However, it is interesting to note that the DAT program described by Minke [49] started at the request of the inmates after a visit from one of the staff members with her dog, suggesting that dogs motivate and generate processes of change in people. Both our results and previously reviewed studies suggest that the motivation and perceived effects of participants in dog-assisted therapy (DAT) programs in the mentally disordered prison population are positive. However, Senneseth et al. [4] established the need for further research on practices that support the recovery process, including ensuring participation in meaningful activities.

## 5. Limitations

Our study has several limitations that warrant consideration. First, the findings cannot be generalized to all inmates with SMD who participate in dog-assisted therapy (DAT) programs, as they are context-specific [38]. Second, the sample size may appear small. However, prior research has shown that justifying sample size in qualitative studies is challenging due to the inherent variability in design, sampling strategy, and data collection methods [51,52]. In this case, the sample size was pragmatically determined by the availability of participants in the DAT program at the Navalcarnero–Madrid IV prison, consistent with other SMD-focused studies [53,54]. Finally, it is necessary to acknowledge that in qualitative research, the researcher and the research process cannot be completely separated, as the researcher and participants mutually influence each other. Therefore, it is essential to explicitly state the relationship between the researcher and the participants [55]. In our study, the therapists responsible for conducting the program sessions (AG-M and AP-G) were part of the research team. Although the researcher conducting the in-depth interviews had no prior relationship with the participants and explicitly clarified that the aim was not to seek positive or negative responses regarding their perception of the program, but rather to learn from their actual experience, a phenomenon of social desirability might have occurred. This could have led participants to refrain from providing negative responses due to the bond formed with the therapists who guided the sessions.

Despite these limitations, this study’s primary strength is its pioneering nature as the first in Spain to explore the experiences of individuals with SMD engaged in a prison-based DAT program. However, conclusions must be drawn cautiously due to the unique design of this intervention, which specifically focused on therapeutic interactions with dogs. By contrast, much of the existing literature examines either visitation programs [15,49] or dog-training initiatives aimed at adoption [25,27,43,46]. This research uniquely contributes to the field by presenting a therapeutic model tailored specifically to a prison population with mental illness in a correctional environment. Additionally, a valuable line of future research could involve incorporating external measures to assess the broader impact of these interactions, such as changes in medication dosages, incidents of disruptive behavior, or other behavioral indicators. These objective measures could provide a more comprehensive understanding of the therapeutic benefits of the program.

## 6. Conclusions

This study provides compelling evidence of the positive impact of dog-assisted therapy (DAT) on the emotional well-being and social connectedness of incarcerated individuals with serious mental disorder (SMD) in the Spanish penitentiary context. It is essential to note that these results are based on self-reported subjective emotional well-being. Participants highlighted that their experience in a prison-based DAT program generated positive emotions, improved mood, and strengthened their sense of connection. The emotional bonds formed with the dog, characterized by unconditional affection and reciprocal trust, helped mitigate feelings of isolation and address the lack of emotional connections typical of prison environments.

The findings also reveal the broader potential of DAT as a rehabilitative tool in correctional settings. For occupational therapy practice, DAT can serve as a meaningful intervention to promote emotional and social rehabilitation. Specifically, it can reduce isolation, foster emotional bonds, and support recovery and resilience among prisoners with SMD. Moreover, the structured integration of DAT into therapeutic programs demonstrates the value of activity-centered approaches, leveraging occupation-based activities to enhance psychosocial rehabilitation.

These results emphasize the need to further investigate and optimize DAT programs to maximize their impact and scalability. Expanding such interventions could enhance rehabilitation efforts in correctional facilities, ultimately improving the quality of life and recovery outcomes for incarcerated individuals with SMD [39].

## Figures and Tables

**Table 1 animals-15-00379-t001:** Semi-structured interview question guide.

Research Areas	Questions
Opening question	How did you find the experience of participating in the dog-assisted therapy program?
Dog-assisted therapy program	Why did you decide to participate in the program? How has the process of participating in the program been from the first to the last session? Has your experience changed or how have you been feeling? What has your participation in the program meant to you? How has the introduction of a dog/animal influenced the intervention? What reflections have you drawn from your interaction with the animal? How has your relationship been with the professionals who have carried out the program? What has been most relevant for you?
Mental health and recovery process	How have you felt during, and after participating in the program? How has it influenced your recovery process? How has it affected your quality of life? How has it affected your health, and specifically your mental health?
Closing questions	Is there any other aspect that you consider important or that you would like to highlight from your participation in the dog-assisted therapy program that we did not cover during the interview? Which one(s)? Finally, could you summarize in one or two sentences your experience of participating in the dog-assisted therapy program?

**Table 2 animals-15-00379-t002:** Sociodemographic and clinical data.

Participants	16 participants (16 men)
Age, years	43.4 ± 11.4
Diagnosis	4 participants: schizophrenia. 3 participants: bipolar disorder. 3 participants: borderline personality disorder. 2 participants: schizoaffective disorder. 2 participants: other psychotic disorder. 1 participant: chronic delusional disorder. 1 participant: unspecified personality disorder
Time to freedom	8 participants: less than 1 year. 3 participants: between 1 and 2 years. 3 participants: more than 2 years. 2 participants: preventive prison, awaiting judgment.
Family support and residential resource upon completion of sentence	10 participants: family support and residential resource. 6 participants: no family support and no residential resource.

## Data Availability

The raw data supporting the conclusions of this article will be made available by the authors on request.

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
