# Peer review of "The Experience of Prisoners with Serious Mental Disorders Participating in a Dog-Assisted Therapy Program: A Qualitative Study"

_animals, 2025, doi:10.3390/ani15030379_

Round 1
Reviewer 1 Report
Comments and Suggestions for Authors
Thank you for the opportunity to read “The experience of prisoners with serious mental disorders participating in a dog-assisted therapy program: a qualitative study.” I found it informative and well designed. I only have a few suggestions:
Line 63: Can start with the name of researchers instead of “the study”
Results: It would be helpful to see example codes
Line 320: I recommend including the citations to the studies referred to here
Line 387: I recommend using dog-assisted intervention instead of dog assisted therapy
Reviewer 2 Report
Comments and Suggestions for Authors
Thank you for the opportunity to read this interesting paper exploring the use of animal-assisted therapy style interventions in the context of carceral spaces. This is an interesting topic, and there is some interesting work here, but the paper needs a little refinement, particularly in connecting to existing literature, being more methodologically explicit, and more strongly identifying its own original contributions rather than concluding by confirming existing work.
The introduction is quite meandering; it is more literature review than introduction. There are opportunities to sharpen this and improve the readability by simplifying and cutting some of the content to get to the matter of AAT quicker. It’s not always clear how the paragraphs necessarily connect and link to one another. The Recovery Model stressed in the first paragraph is never mentioned again, for example. The content is fine, but needs to be more organised with a more consistent narrative thread to help guide a reader, rather than jumping from topic to topic.
The review of AAI and DAT could be expanded. It is very short. There is a lot of research on the use of animal therapy in prisons that is not cited here. Furst 2006; Mercer et al 2015; Strimple 2003; Fournier et al 2007; Mulcahy & McLaughlin 2013, etc etc. A more detailed and specific review is really needed to avoid feeling rushed.
Section 2.2 needs more. It is fine to say more detail is in the supplementary material, but a sentence or two should at least be given here. It is so important for the analysis and results that follow that it really should be integrated into the paper. A reader cannot understand what follows without it. I hope the editors will allow some additional word count flexibility to enable this.
Rather than simply stating that ethical approval was given by a governing body, it would be good to state and discuss what the ethical issues at stake were. Similarly for the animal welfare.
The authors need to provide a definition of what they mean by ‘SMD’ – what is included in this?
The analysis needs a better grounding in thematic analysis. For example, the authors state “the text fragments that provided relevant information for the study were identified” – but how was ‘relevance’ conceptualised and chosen. This is an active decision, not something that is inherent in the data. What particular ‘lens’ were the authors applying to decide ‘relevance’. Remember, as Braun and Clarke remind us… “Themes do not passively emerge from either data or coding; they are not ‘in’ the data, waiting to be identified and retrieved by the researcher. Themes are creative and interpretive stories about the data, produced at the intersection of the researcher’s theoretical assumptions, their analytic resources and skill, and the data themselves.”
The authors are at risk of simply naming processes involved in qualitative analysis, rather than describing how these were specifically utilised and applied within this specific study. E.g. “Triangulation of researchers and data collection instruments was performed during the analysis, together with member checking, complete description of the study, a reflexivity process and description of tree coding from the narratives to the final themes.” – what did all of this look like in practice? What did member-checking reveal or change? What did applying Lincoln and Guba do to the authors data/practices? More descriptive specificity of the actual _process_ of analysis is needed, rather than just naming stages.
There is a need for consistent language – subtheme or subtopic? It would also be good to see the themes/subthemes a little more balanced. Some are very short compared to others. The theme headings could also be a bit more creative, some (the decision to participate) seem more categories than themes.
Subtopic 1.2 is good as it relates the theme particularly back to the context of being in prison and why this is important. 1.1 could make this connection a little more specifically as it feels a little too broad at present.
Broadly, the empirical data is also very positive and affirmative. Where there no negatives? Who was excluded? Was the AAT ever unsuccessful? There are opportunities for more criticality.
Again, consistent terminology is needed. The first half of the paper refers to DAT, this suddenly switches to DAP in the discussion.
The discussion feels more like a literature review. It needs to explicitly link back to the data a little more. It also feels far too confirmatory – what is particular and new from this study that is not already present in the literature?
The conclusion is a little strong, given what data has been reported – I think recognizing that this is ‘self-reported subjective emotional well-being’ is important here.
The original aim specified the ‘Spanish context’ very strongly, but this is lost in the results, discussion, and conclusion. What is particular about the Spanish experience here? The latter half of the paper could be read as a study of anywhere.
The authors conclude by arguing that much existing literature examines visitation programs – however, it is not clear how their work is different? Again, without a wider account of the literature around AAT in prisons, this final claim to ‘uniqueness’ falls short.
The paper could also be more considered and reflective about how language (“mental disorders”) can be stigmatising and whether such medicalised terminology is correct and useful here.
Reviewer 3 Report
Comments and Suggestions for Authors
This is an interesting addition to the literature on dog-assisted therapy programs aimed at prison populations. It is unique in that the subject population consisted of prisoners with serious mental disorders. However, one concern is that there is little information provided about the diagnoses of the subjects. Table 2 describes 4 of the 16 subjects as having a diagnosis of schizophrenia. No details are provided about the remaining 12 men, nor is there any discussion of possible effect of diagnosis on the nature of the interactions of the subject with the therapy dog.
The supplemental material provides few details of the kinds of activities with the dog that were available to the subjects beyond labels (grooming, walking, dog sniffing training, contact and games). The protocol for each of these activities should be provided in the supplement. In addition, it would be useful to see how often each of these were selected by participants and, if possible, if the specific activities chosen had any impact on responses in the interviews.
The supplement mentions the use of only one dog (Ara) yet the discussion of the results repeatedly mentions “bonding with the dogs”. This should be clarified. Did all the interactions involve only one dog and was it always the same dog?
It would be worth noting if there were any other external measures of the potential impact of the interactions with the dog, as have been noted in other studies. This might include changes in necessary dosages of medication, changes in reports of incidents such as disruptive behavior, etc.
Round 2
Reviewer 2 Report
Comments and Suggestions for Authors
The manuscript has been much improved, however there is still work to be done. Some of my previous queries and concerns have not been fully addressed, whilst other edits have raised new challenges.
On the previous version, I asked that the rather than simply stating that ethical approval was given by a governing body, the authors discuss what the ethical and animal welfare issues at stake were. This has not been completed, instead we are given even more information on the governance and guidelines and requirements – this is not interesting or helpful, it needs to be specific to this study and what was done. _What_ are the ethical issues of doing research with incarcerated people? How is consent managed in such a context? What did the researchers do to manage their relationships with participants? What are the ethics of working with people with SMD? How was informed consent obtained?
On the previous version, I expressed a concern that the methods of analysis were not described in enough detail. This unfortunately remains the case. The authors now state that thematic relevance was “defined in relation to the chosen theoretical framework” – what is the theoretical framework here? No theoretical framework has been introduced or described.
In their response the researchers mention they “have not identified any negative effects/comments/reflections. Therefore, the data reflect only the perceptions and positive outcomes reported by them.” This is perhaps a little concerning regarding the methods of data collection, the depth and richness of the interviews, and the potential for having biased or perhaps lead interviewees into reporting certain aspects. How was this managed? This lack of negative reportage still needs to be accounted for, reflected on and discussed in the limitations section. Why did people perhaps not feel comfortable or able to report downsides - this is particularly important given the carceral context! Perhaps people felt that reporting any negatives would result in the intervention being taken away. This time of criticality and reflectiveness would be helpful across the whole manuscript.
